# Evaluating the predictive accuracy of cognitive screeners BAMCOG and MoCA in identifying postoperative delirium risk in aortic valve replacement patients: A cohort study

Mariska E. te Pas[1,2]*, Sophie Adelaars[1,2], R. Arthur Bouwman[1,2], Roy P.C. Kessels[3,4,5], Marcel G.M. Olde Rikkert[6], Daan van de Kerkhof[7], Erwin Oosterbos[1], Marc P. Buise[8]

1 Department of Anesthesiology, Catharina Hospital Eindhoven, Eindhoven, The Netherlands, 2 Department of Electrical Engineering, Signal Processing Systems, Eindhoven University of Technology, Eindhoven, The Netherlands, 3 Donders Institute for Brain, Cognition and Behaviour, Radboud University Nijmegen, Nijmegen, Gelderland, The Netherlands, 4 Department of Medical Psychology & Radboudumc Alzheimer Center, Radboud University Medical Center Nijmegen, Nijmegen, The Netherlands, 5 Vincent van Gogh Institute for Psychiatry, Venray, The Netherlands, 6 Department of Geriatrics, Radboudumc Alzheimer Center, Radboud University Medical Center, Nijmegen, Georgia, The Netherlands, 7 Clinical Laboratory, Catharina Hospital Eindhoven, Eindhoven, The Netherlands, 8 Department of Anesthesiology, Maastricht UMC+, Maastricht, The Netherlands

* mariska.t.pas@catharinaziekenhuis.nl

## Abstract

Postoperative delirium (POD) and postoperative encephalopathy (POE) are common complications in older adults undergoing aortic valve replacement (AVR), yet the predictive accuracy of cognitive screening tools remains uncertain. In this prospective cohort study, 50 patients aged 65 years and older scheduled for AVR between January and October 2022 underwent preoperative assessment with the Brain Aging Monitor Cognitive Assessment (BAMCOG) and Montreal Cognitive Assessment (MoCA). Postoperatively, POD was evaluated with the Delirium Observation Screening (DOS) scale and POE with electroencephalography (EEG). BAMCOG and MoCA showed poor accuracy in predicting POE, with AUROC values of 0.67 and 0.59 respectively, but BAMCOG demonstrated good accuracy for POD prediction (AUROC 0.85) compared with MoCA (AUROC 0.53). Higher BAMCOG scores were significantly associated with reduced POD incidence, with each 10% increase in score lowering the risk by 16%. These findings suggest that BAMCOG may be a valuable preoperative screening tool for POD, though larger studies are needed to confirm its clinical utility and establish optimal cutoff values.

## Author summary

In our study, we explored how well simple cognitive tests can help identify older adults at risk of developing confusion after heart valve surgery. This confusion,

**Data availability statement:** The datasets generated and/or analyzed during the current study are available in the Zenodo repository, DOI: 10.5281/zenodo.15241438, and can be accessed at https://zenodo.org/records/16810812.

**Funding:** The author(s) received no specific funding for this work.

**Competing interests:** The authors have declared that no competing interests exist.

known as postoperative delirium, is common and can cause longer hospital stays, reduced quality of life, and even higher mortality. One of the most important risk factors for delirium is preoperative cognitive dysfunction, meaning subtle problems with memory, attention, or thinking that may not yet be obvious in daily life. To address this, we tested two screening tools before surgery: the Montreal Cognitive Assessment (MoCA), which is widely used in hospitals, and the Brain Aging Monitor Cognitive Assessment (BAMCOG), a tablet-based set of short games designed to measure different aspects of thinking. We found that BAMCOG performed much better than MoCA in predicting who would later develop delirium, with higher BAMCOG scores linked to a lower risk. Because BAMCOG is game-like and self-administered, it may also be more engaging and less stressful for patients. Larger studies are needed to confirm these promising findings.

## 1. Introduction

Postoperative delirium (POD), a state of acute confusion and altered mental function following surgery [1], is a serious complication linked to adverse outcomes including increased mortality risk, long-term cognitive decline, reduced quality of life, prolonged hospital stay and increased healthcare costs. [2–4] Addressing and managing POD remains a significant challenge, mainly because many healthcare professionals are not fully aware of its occurrence and because of the absence of effective treatments. [5,6] Therefore, the prevention of POD through targeted preoperative interventions-such as lifestyle and nutritional optimization, cognitive prehabilitation, and physical training remains key. [7,8] Identifying patients at risk is a critical first step, making preoperative screening for cognitive dysfunction essential due to its significance as a major risk factor for POD. [9]

Detecting neurodegenerative cognitive dysfunction in its early stages, referred to as Mild Cognitive Impairment (MCI), is complicated due to the variability in symptoms during the initial onset, which depends on factors like aetiology and cognitive reserve. [10] Additionally, limited time within the healthcare system and lack of participant motivation for assessment with traditional cognitive screeners or tests present obstacles to early detection, potentially impacting the quality of screening data. [11]

Introducing gamification into cognitive screening possibly offers promising, self-administrable an alternative to traditional methods. Engaging users through challenging puzzle minigames could enhance assessment accuracy and participant involvement. In this study, we evaluate the potential of such a gamified cognitive screener - the Brain Aging Monitor – Cognitive Assessment (BAMCOG), alongside the traditional Montreal Cognitive Assessment (MoCA), in identifying patients at risk for POD following Aortic Valve Replacement (AVR) surgery. [12,13]

## 2. Materials and methods

### 2.1 Ethics statement

This study is based on an observational clinical trial registered at ClinicalTrials.gov (NCT05209555) and approved by the Medical Research Ethics Committee (MEC-U,

Nieuwegein, The Netherlands; approval number A21.313/R21.054) and the Institutional Review Board of Catharina Hospital (Eindhoven, The Netherlands; approval granted). Written informed consent was obtained from all participants prior to inclusion. The study was conducted in accordance with the principles of the Declaration of Helsinki (Fortaleza, Brazil, October 2013) and with applicable Dutch law.

## 2.2. Participants and sample size

A cohort study was performed in the Catharina Hospital in Eindhoven, the Netherlands, a tertiary referral hospital, providing cardiac surgery. The sample size of 50 patients is determined by assuming a 50% incidence of POD in this patient population. This sample suffices to achieve a significant difference compared to an AUROC of 0.5 (for MoCA) and a minimum BAMCOG Area Under the Receiver Operation Curve (AUROC) of 0.72 with $\alpha = 0.05$ and a power $(1-\beta)$ of 0.8. To account for a 20% loss to follow-up, sixty patients scheduled for open aortic valve replacement (AVR) surgery were enrolled in this study between January and October 2022. Inclusion criteria comprised individuals aged 65 years or older, scheduled for AVR surgery, proficient in Dutch, and capable of engaging in tablet-based gaming. Exclusion criteria encompassed documented learning disorders (e.g., dyslexia, dyscalculia, non-verbal disorder and language development disorder), psychiatric disorders, dementia, use of lithium or clozapine, presence of a pacemaker, defibrillator or neurostimulator, and three or more days of postoperative sedation. Ten patients were excluded during the study period due to the transition to an alternative surgical procedure (6 cases), absence of cardiopulmonary bypass during surgery (1 case), and refusal to participate (3 cases), resulting in a final analysis cohort of fifty patients.

## 2.3. Materials

### 2.3.1 MoCA.
The Montreal Cognitive Assessment scale (MoCA) is a one-page, 30-point test developed by Nasreddine et al., that helps healthcare professionals detect subtle signs of cognitive impairment. [14] The assessment can be conducted within a 10–15 minute timeframe, evaluating short-term memory, visuospatial abilities, executive functions, attention, concentration, working memory, language, and orientation in time and place. [15] The sensitivity and specificity ranges to detect Mild Cognitive Impairment in an elderly memory clinic population with the MoCA are 81–93% and 74–89%, respectively. [16–19] In this study, different versions of the MoCA (Dutch V8.1, 7.2, and 7.3, respectively) were used at various measurement points, to minimize potential learning effects associated with repeated cognitive testing. Specifically, version 8.1 was administered preoperatively and again on postoperative day 7, while version 7.2 was used on postoperative day 1 and version 7.3 on postoperative day 3.

### 2.3.2 BAMCOG.
The Brain Aging Monitor for Cognition (BAMCOG) was originally developed as an online self-monitor for cognitive functioning for use on a laptop or personal computer. [13] For this study, the BAMCOG was newly programmed for use as an application on a tablet (Game Development Software Unity®). [20]

The BAMCOG application comprises three games. The initial game, named "Groceries," focuses on working memory. Participants are shown a grocery list on the screen, and after 1 second, a conveyor belt with groceries activates. Participants must then select the correct products. The maximum score for this game is 590 points. The second game, "Memory," revolves around visuospatial short-term memory. Visual patterns are presented in a 5 × 5 cloud matrix, and participants are tasked with reproducing the pattern in the exact order it initially appeared on the screen. The maximum score for this game is 620 points. The third game, "Connect the Line," pertains to executive function and planning. Participants are initially presented with a scrambled path, and their objective is to unscramble the path by sliding columns and rows in the correct order, allowing their pawn to move unobstructed from start to finish. The maximum score for this game is 80 points. Each game begins with a practice level.

Because of the wide range of scores in the different BAMCOG games, normalized scores were calculated. The overall score is represented as the average of the percentages achieved in the three distinct games. For example, a patient obtained scores of 420, 350, and 40 points in Groceries, Memory, and Connect the Line, respectively. These translate to

percentages of 71% (420 out of 590), 56% (350 out of 620), and 50% (40 out of 80). The average percentage of these values is 59%, serving as the total score for this patient.

**2.3.3. DeltaScan.** The DeltaScan Brain State Monitor is a bedside EEG monitor which determines within a few minutes if a patient suffers from Acute Encephalopathy (AE) or not. [21] AE is a term referring to a state of generalized brain dysfunction that may or may not include features of delirium. EEG recording will be made using a strip with EEG electrodes that will be mounted to the head using self-adhesive gel. A score of 1–2 means a normal EEG, whereas a score of 3–5 means an EEG with signs of AE. [22]

**2.3.4 DOS.** The DOS consists of 13 questions about early symptoms of delirium that nurses could observe during regular care. Each item could be rated with 0 (normal) or 1 (abnormal), with a scoring range from 0 to 13. The cut off value of 3 points or more indicates delirium. [23]

## 2.4. Procedure and analysis

Informed consent was obtained, and the initial baseline measurements for DeltaScan, MoCA (Dutch V8.1) and BAMCOG were conducted at the day before surgery. Postoperatively, MoCA and BAMCOG assessments were repeated on the first, third, and seventh day, once daily using alternate forms (MoCA Dutch V8.1, 7.2 and 7.3, respectively). Due to the variable course of delirium, DeltaScan and DOS (Delirium Observation Scale) were measured twice a day postoperatively, also on the first, third and seventh day. Fig 1 summarizes the measurements performed.

This paper addresses two research questions: the main is whether postoperative delirium (POD) and postoperative encephalopathy (POE) can be predicted using preoperative MoCA and BAMCOG scores, where POD and POE are defined as the occurrence, at least once, of a Delirium Observation Screening (DOS) score greater than 3 and a DeltaScan score greater than 3, respectively. The second question examines differences between delirious and non-delirious patients based on factors such as age, co-morbidities, pre- and perioperative medication use, alcohol consumption, and duration of extracorporeal perfusion.

| | Day | MoCA | BAMCOG | DOS Morning | DOS Afternoon | DeltaScan Morning | DeltaScan Afternoon |
|---|---|---|---|---|---|---|---|
| Preoperative visit and informed consent | 1 | | | | | | |
| Day 1 before surgery | 14 | 🔍 | 🔍 | 🔍 | | 🔍 | |
| Day of surgery | 15 | | | | | | |
| Postoperative day 1 | 16 | 🔍 | 🔍 | 🔍 | 🔍 | 🔍 | 🔍 |
| Postoperative day 3 | 18 | 🔍 | 🔍 | 🔍 | 🔍 | 🔍 | 🔍 |
| Postoperative day 7 | 22 | 🔍 | 🔍 | 🔍 | 🔍 | 🔍 | 🔍 |

**Fig 1. Summary measurements during study.**

Descriptive data are presented as mean and standard deviation (SD), median and interquartile range (IQR) where appropriate. AUROC calculations with 95% confidence intervals of the ROC-curves were created using RStudio. A logistic regression analysis was conducted (using RStudio) to examine the association between BAMCOG scores and POD (measured by DOS). To classify the level of education, the Dutch Verhage scale was used. [24]

## 3. Results

The study sample consisted of 50 participants, including 9 women, with a mean age of 72.7 years (SD 4.2; range 66–80). 54% of the patients were classified as ASA 3, while the remaining patients were classified as ASA 4. [25] None of the participants had a DeltaScan score greater than or equal to 3 on the day prior to the surgery. Other baseline characteristics are shown in Table 1. Table 2 presents the incidences of POD (N = 14, 28%) and POE (N = 38, 76%), as well as perioperative differences between patients with and without POD, and those with and without POE. ROC curves show that preoperative MoCA and BAMCOG poorly predict POE (AUROC of 0.59, 95% CI: 0.41-0.77 and 0.67, 95% CI: 0.50-0.85, respectively) (Fig 2). Fig 3 shows that preoperative MoCA scores had no predictive value for POD (AUROC 0.53, 95% CI: 0.33-0.73), while prediction of POD with BAMCOG was good (AUROC 0.85, 95% CI: 0.72-0.98). Logistic regression showed that higher preoperative BAMCOG scores are highly associated with a lower incidence of POD (OR 0.984, 95%CI: 0.978 – 0.991, P < 0.001) (Fig 4). Specifically, a 10% increase in BAMCOG score corresponds to a 16% reduction in the likelihood of developing POD (95% CI: 9%-22%). Adding the predictive variables 'age' and 'educational level' to the logistic regression model did not result in statistical significance (P = 0.65 and P = 0.93, respectively). Table 3 provides a summary of the predictive performance and logistic regression results. Although no statistically significant perioperative

**Table 1. Baseline characteristics.**

| Characteristic | N = 50 |
| --- | --- |
| Mean age (SD) | 72.7 (4.2) |
| 65–69 | 14 (28) |
| 70–74 | 17 (34) |
| 75–79 | 18 (36) |
| 80–84 | 1 (2) |
| Male (%) | 41 (82) |
| ASA[1] 3 (%) | 27 (54) |
| ASA[1] 4 (%) | 23 (46) |
| Education level[2] | |
| N Low education level (%) | 14 (28) |
| N Average education level (%) | 21 (42) |
| N High education level (%) | 12 (24) |
| N Not available education level (%) | 3 (6) |
| BMI[3] in Kg/m2 (SD) | 28.0 (4.1) |
| UA[4] Alcohol use per day, preoperative (range) | 1.0 (0-14) |
| Amount of difference medicines preoperative (SD) | 4.9 (2.7) |
| Dexamethasone during surgery (%) | 34 (68) |

[1]ASA Classification = American Society of Anesthesiologists.

[2]Low educational level = 1–4, average educational level = 5, high educational level = 6–7, according to Verhage Classification [24].

[3]BMI = Body Mass Index.

[4]UA = Units of Alcohol.

**Table 2. Perioperative differences between delirious and non-delirious patients.**

| Characteristic | Postoperative Encephalopathy (N=38) | No Postoperative Encephalopathy (N=12) | Postoperative Delirium (N=14) | No Postoperative Delirium (N=36) |
|---|---|---|---|---|
| Mean age (years) | 72.9 (0.7) | 72.1 (1.2) | 73.2 (4.3) | 72.5 (4.2) |
| Mean Body Mass Index (BMI) (Kg/m2) | 27.7 (4.0) | 28.9 (4.3) | 27.8 (5.2) | 28.0 (3.6) |
| Mean surgery time (min) | 189.8 (47.6) | 204.1 (36.1) | 196.2 (61.8) | 192.1 (37.9) |
| Mean perfusion time (min) | 99.3 (35.0) | 88.8 (17.8) | 112.1 (49.8) | 90.8 (19.3) |
| Mean alcohol use a day preoperative (UA)[1] | 1.0 (0.4) | 0.8 (0.2) | 0.7 (0.8) | 1.1 (2.3) |
| Mean of medication preoperative (number) | 4.8 (3.0) | 5.1 (1.4) | 4.5 (2.9) | 5.0 (2.9) |
| Mean amount of propofol during operation (g) | 1.7 (0.23) | 1.9 (0.6) | 1.8 (5.3) | 1.8 (5.0) |
| Mean amount of alfentanil during operation (mg) | 1.4 (0.3) | 1.4 (0.3) | 1.4 (0.4) | 1.4 (0.3) |
| Dexamethasone during surgery (%) | 25 (66) | 9 (75) | 9 (64) | 25 (69) |
| Mean length on Intensive Care Unit (days) | 1.1 (1.4) | 0.7 (0.8) | 1.6 (2.0) | 0.8 (0.7) |
| Mean length of stay overall (days) | 5.7 (4.4) | 4.8 (2.3) | 6.7 (6.1) | 5.03 (2.8) |

[1]UA=Units of Alcohol

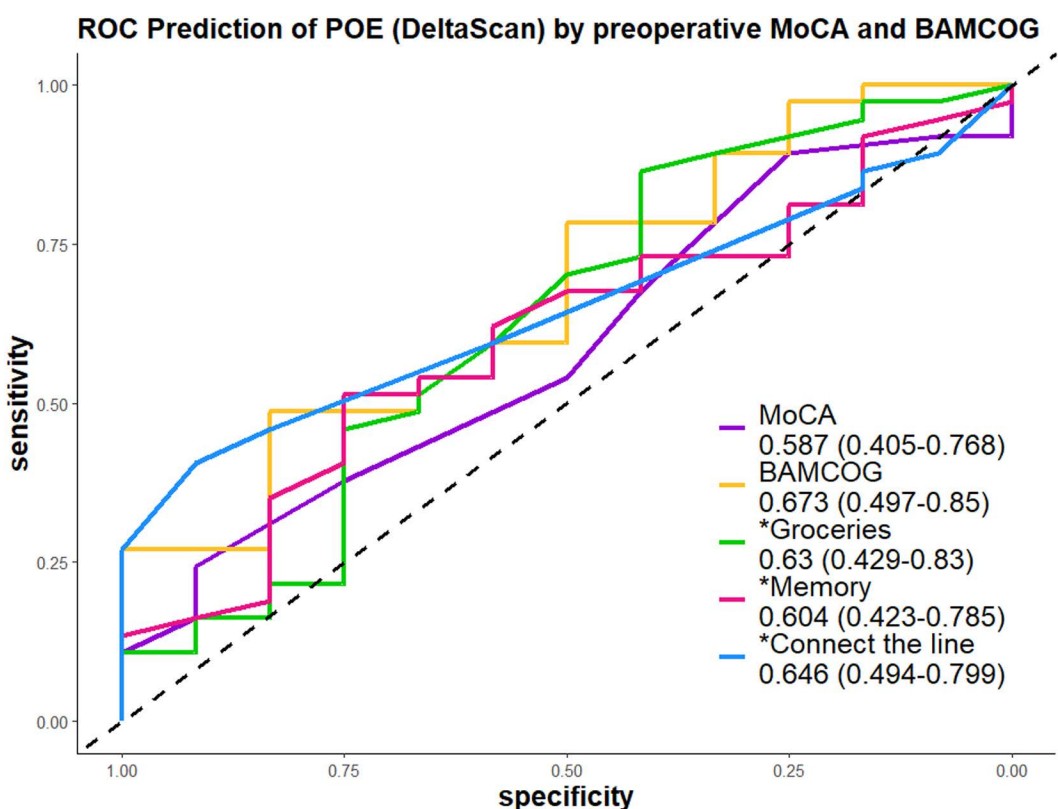

**Fig 2. Receiver Operating Characteristic Curve for Predicting Postoperative Encephalopathy (POE, determined by DeltaScan) Using Preoperative MoCA and Normalized BAMCOG Scores.**

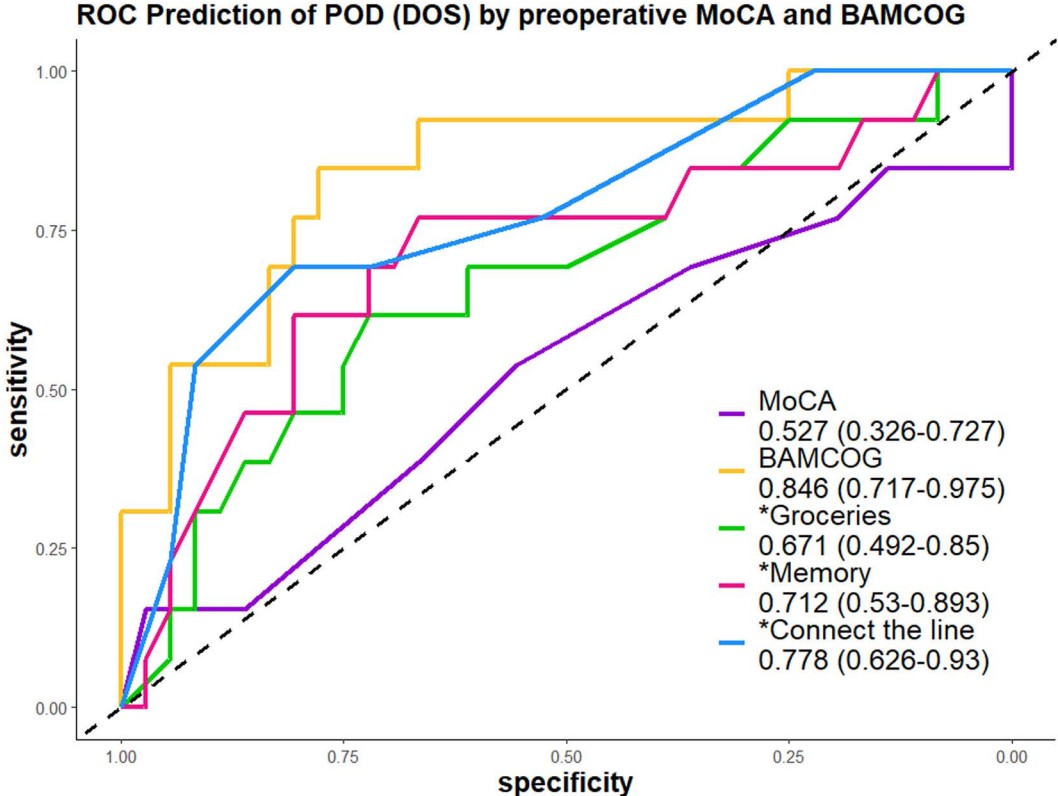

**Fig 3. Receiver Operating Characteristic Curve for Predicting Postoperative Delirium (POD, determined by Delirium Observation Score (DOS)) Using Preoperative MoCA and Normalized BAMCOG Scores.**

differences were observed between patients with and without POD or POE, those who developed POD or POE had longer median ICU and total hospital stays compared to patients without these conditions (Table 2).

## 4. Discussion

### 4.1. Results and interpretation

This study suggests that the BAMCOG poorly predicts postoperative POE, but demonstrates a good predictive performance for POD classified with DOS. In contrast, the MoCA poorly predicts both POE and POD. Logistic regression analysis indicated that higher preoperative BAMCOG scores are significantly associated with a decreased incidence of POD measured by DOS. Another interesting result of this study is the average reduction in hospital stay by 1 to 1.5 days for patients without POE and POD, highlighting the importance of identifying high-risk individuals to improve perioperative care.

This study provides, to our knowledge, the first evaluation of the predictive value of both the BAMCOG and the MoCA POD and POE. However, our findings regarding BAMCOG's ability to predict POD add to previous studies, such as Dworkin et al. [26] and Segernäs et al. [27], which highlighted the validity of cognitive screeners (Mini-Cog and MMSE, respectively) in predicting POD after cardiac surgery. Also other predictive approaches such as a machine learning model of Nagata et al. have shown promise. [28] However, unlike these models and conventional tests, our study shows potential advantages of tablet-based assessments, such as reducing stress and making the process feel less like a traditional test, thereby minimizing performance anxiety and producing more valid cognitive responses. Consistent with this, another key advantage of our tablet-based games

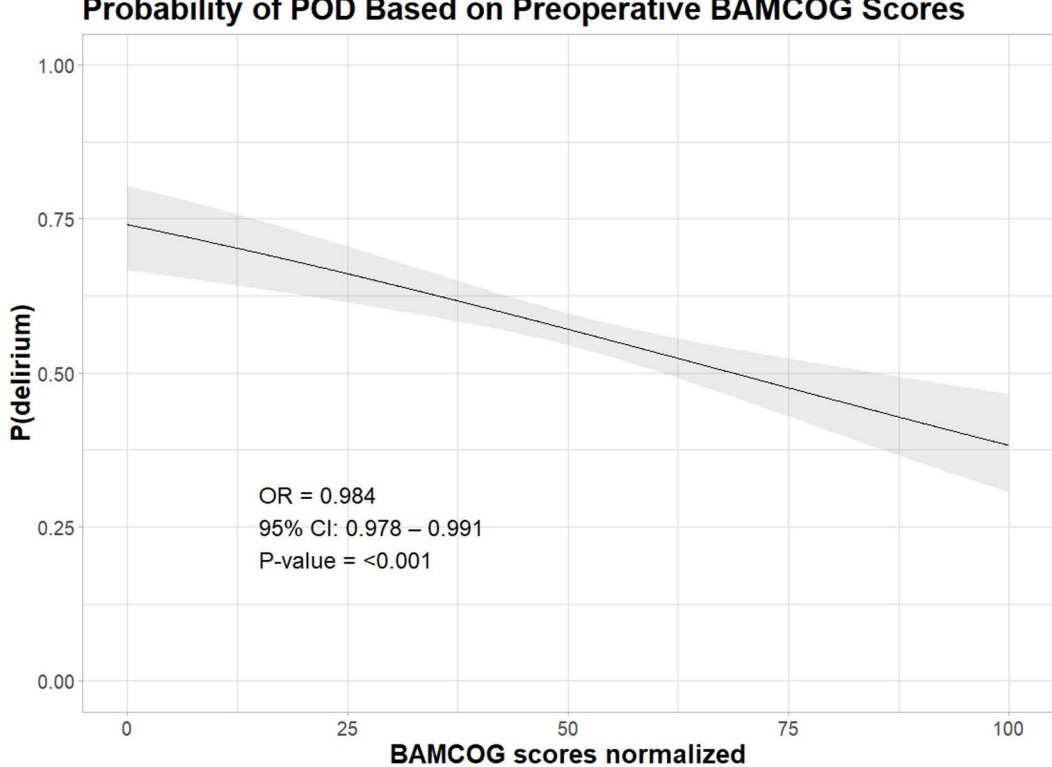

**Fig 4. Logistic regression association preoperative BAMCOG scores and Postoperative Delirium (POD).**

**Table 3. Predictive performance of preoperative MoCA and BAMCOG scores for POE and POD.**

| Test | Outcome | Measure | Result |
|---|---|---|---|
| **MoCA** | POE | AUROC (95% CI) | 0.587 (0.405–0.768) |
| **BAMCOG** | POE | AUROC (95% CI) | 0.673 (0.497–0.85) |
| **Groceries** | POE | AUROC (95% CI) | 0.63 (0.429–0.83) |
| **Memory** | POE | AUROC (95% CI) | 0.604 (0.423–0.785) |
| **Connect the line** | POE | AUROC (95% CI) | 0.646 (0.494–0.799) |
| **MoCA** | POD | AUROC (95% CI) | 0.527 (0.326–0.727) |
| **BAMCOG** | POD | AUROC (95% CI) | 0.846 (0.717–0.975) |
| **Groceries** | POD | AUROC (95% CI) | 0.671 (0.492–0.85) |
| **Memory** | POD | AUROC (95% CI) | 0.712 (0.53–0.893) |
| **Connect the line** | POD | AUROC (95% CI) | 0.778 (0.626–0.93) |
| **BAMCOG** logistic regression | POD | OR (95% CI) | 0.984 (0.978–0.991), P < 0.001 |
| **BAMCOG** 10% increase | POD | Risk Reduction (95% CI) | 16% (9%–22%) |

*POE = Postoperative Encephalopathy. POD = Postoperative Delirium. AUROC = Area Under the Receiver Operating Curve. CI = Confidence Intervals. OR = Odds Ratio.*

is that they are self-administered, allowing patients to play in a private setting without the researcher's active involvement or questioning. In contrast, tests like the MoCA may induce stress due to real-time observation and supervision by the researcher or clinician. Lastly, these previous studies did not or explore the potential benefits of gamified systems, as we do.

## 4.2. Role of gamification

Gamification could explain also the difference in predictive power between BAMCOG and MoCA, as it may elicit more ecologically valid cognitive responses compared to traditional methods. While research on gamified preoperative cognitive screening and its predictive value for POD remains limited, extensive studies in other healthcare domains have shown promising outcomes for gamification. For instance, a meta-analysis by Vermeir et al. [29] revealed that gamified cognitive tasks significantly enhance motivation and engagement compared to conventional tasks, underscoring the potential benefits of gamification, which we revealed in this study. Additionally, research on gamification in mental health and health behaviour has demonstrated substantial positive effects. A systematic review [28] identified seven key benefits of gamification in health and well-being, with the primary advantage being the enhancement of intrinsic motivation. [30] Furthermore, another review [31] on gamification in mental health highlighted its potential to increase patient engagement and motivation, helping individuals achieve therapeutic goals, such as promoting mental wellness and alleviating psychological symptoms. Supporting these findings, our research group reported a good System Usability Scale (SUS) score of 79.7 for BAMCOG usability, potentially attributed due to the tool's game-like and fun elements. (manuscript submitted)

## 4.3. Gold standard delirium diagnosis

Our results indicate varying predictive values depending on the diagnostic method used for delirium. While the gold standard for diagnosing delirium involves assessment by a delirium expert using Diagnostic and Statistical Manual of mental disorders (DSM-5-TR) criteria, in practice, this process is challenging to implement consistently in a busy hospital setting. Tools such as the DOS, with reported sensitivity and specificity of 90% and 91% [32,33], serve as practical alternatives when expert evaluation is not immediately available. However, previous research indicates significant interobserver variability in diagnosing delirium using these tools. [34,35] The inherent subjectivity of DOS scores highlights the need for more objective methods of delirium detection.

To address this, we incorporated the DeltaScan into our study, aiming to enhance delirium detection by integrating multiple diagnostic approaches. However, our results reveal that POE was observed in 38 patients (76%), whereas only 14 patients (28%) exhibited the phenotype of postoperative delirium as determined by DOS. An interpretation for these results is that individuals experiencing POE may be in a pre-stage of delirium that is not inherently linked to preoperative cognitive function. However, the transformation into delirium or its absence appears to be associated with the individual's preoperative cognitive function.

## 4.4. Limitations

A key limitation of this study is the lack of a definitive gold standard for diagnosing delirium. [36] While the DSM-5-TR and ICD are reference standards, their use is limited to clinical settings with trained professionals, reducing practicality. [37] We used DeltaScan and DOS to address this, but evidence supporting DeltaScan's efficacy, particularly for patients with pre-operative cognitive impairments, remains limited. Additionally, DeltaScan is not yet considered a gold standard for delirium detection. The same applies to the (DOS), which – although widely used in clinical practice – is a screening tool rather than a diagnostic standard, and may miss hypoactive or atypical forms of delirium. [38,39]

Another limitation is the lack of evidence regarding BAMCOG's predictive accuracy for mild cognitive impairment (MCI). Although our prior research (manuscript submitted) showed moderate validity with MoCA for detecting MCI, BAMCOG was initially designed for online self-monitoring rather than as a tablet-based cognitive screener. [13] However, this study highlights the potential of BAMCOG to possibly identify patients at risk for POD, regardless of its accuracy in detecting MCI.

A further consideration concerns the potential influence of digital literacy when using a game-based, tablet-delivered cognitive screener such as BAMCOG. Older adults may vary in their familiarity and comfort with digital devices, which could theoretically affect performance. In our analysis, age was included as a proxy for digital proficiency, but it did not significantly affect the association between BAMCOG scores and POD risk. This suggests that the predictive value of

BAMCOG is not merely driven by differences in digital skills, but rather reflects cognitive functioning. Importantly, previous research by Wu et al. [40] and Bayer et al. [41] has shown that tablet-based cognitive tests and digital cognitive assessment tools are valid, feasible, and usable among older adults, supporting the potential of such tools in clinical screening contexts. Nonetheless, future research should consider incorporating more direct measures of digital literacy to further examine its potential impact.

Lastly, the causes of POD after cardiac surgery, such as reperfusion ischemia, non-pulsatile flow due to cardiopulmonary bypass (CBP), systemic inflammation and cerebral hyperthermia, remain debated. [42,43] Logically, the effects of these perioperative mechanisms would manifest in the initial postoperative days. In our study, most cases of delirium and encephalopathy occurred within the first three postoperative days, likely linked to surgery-specific factors. Similar trends have been observed in non-cardiac surgery studies, such as that of Iamaroon et al. [44]

### 4.5. Recommendations

To establish BAMCOG as a reliable preoperative cognitive screening tool for POD prediction, further studies are needed to validate its efficacy. This involves setting appropriate cutoff scores for BAMCOG and assessing how factors like age and education influence results. A sample size of approximately 230 patients would allow for a reasonable estimation of both sensitivity and specificity of a diagnostic tool like DeltaScan, assuming a delirium prevalence of 20%, an expected sensitivity of 80%, and a confidence level of 95% with a margin of error of approximately 10%. Additionally, clinical validation should include confirming delirium diagnoses with experts like geriatricians or psychiatrists, despite the challenges of time and cost. Future research should also investigate BAMCOG's role in detecting mild cognitive impairment (MCI), which was not a primary focus of this study. Finally, a study that conducts preoperative cognitive screening further in advance of surgery and evaluates subsequent prehabilitation programs will help validate our hypothesis that prehabilitation can reduce the incidence of POD.

### 4.6. Conclusions

Our findings indicate a limited predictive performance of the MoCA for anticipating postoperative Encephalopathy and Delirium. However, they indicate a good predictive performance of the BAMCOG for predicting postoperative delirium as detected by DOS. This suggests that preoperative cognitive screening with BAMCOG could hold promise in identifying patients who could benefit from preoperative prehabilitation and lifestyle interventions aimed at reducing the risk of postoperative delirium. While BAMCOG lacks clinical validation for diagnosing MCI, further research is necessary to establish its clinical utility as a preoperative cognitive screener. Future studies with larger sample sizes across diverse surgical patient populations and using optimal gold-standard delirium diagnoses are needed. Nonetheless, our study highlights the potential benefits of incorporating gamification in preoperative cognitive screening, opening new avenues for enhancing patient care and outcomes.

### Author contributions

**Conceptualization:** Mariska te Pas, Sophie Adelaars, R. Arthur Bouwman, Marc P. Buise.

**Formal analysis:** Mariska te Pas.

**Investigation:** Mariska te Pas, Sophie Adelaars, Erwin Oosterbos.

**Methodology:** Mariska te Pas, Sophie Adelaars, R. Arthur Bouwman, Marc P. Buise.

**Supervision:** R. Arthur Bouwman, Marc P. Buise.

**Writing – original draft:** Mariska te Pas, Sophie Adelaars.

**Writing – review & editing:** R. Arthur Bouwman, Roy P.C. Kessels, Marcel G.M. Olde Rikkert, Daan van de Kerkhof, Marc P. Buise.

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
