## [Decision Letter · Decision Letter 0]

1 Aug 2025

Response to Reviewers
Revised Manuscript with Track Changes
Manuscript
**Journal Requirements:**
**Additional Editor Comments (if provided):**

1. Should the abbreviation for postoperative encephalopathy be POE instead of the PAE currently used in the manuscript?

2. The first line of the results is, in my view, confusing. Currently it reads: "The sample comprised 9 women...". I initially thought this meant that there were a total of 9 study participants, whereas actually there were 50 participants, 9 of whom were women. Please consider re-phrasing this.

3. Figure 1: Should the title read "ROC Prediction of PAE" instead of "ROC Prediction of AE"?

4. Figures 1 and 2: Whilst I appreciate the shaded areas provide information on confidence intervals (I presume), in this particular case I would suggest you consider leaving out the shaded areas. In my view they add little information as it is difficult to discern the individual areas, they make it more difficult to see the ROC curves, and they may even be misleading because there are more shading colours than ROC curves (due to merging colours in the overlapping areas) which places emphasis on the overlapping areas rather than the individual areas.

5. Section 2.2.1 states that different versions of MoCA were used at various measurement points. Please could you clarify whether the different measurement points refer to: (i) different days (e.g. day 1, 14, 15, 16, ... as illustrated in Table 1); or (ii) changes during the overall study (e.g. the first 25 participants used one version, and the next 25 participants used a different version). Please also briefly state / discuss whether or not the use of different versions could impact the results and conclusion.

6. In the discussion please discuss any previous work reporting the accuracy of MoCA and BAMCOG in predicting POD and PAE.

**Reviewers' Comments:**

**Comments to the Author**

1. Does this manuscript meet PLOS Digital Health’s publication criteria?

Reviewer #1: Yes

2. Has the statistical analysis been performed appropriately and rigorously?

Reviewer #1: Yes

3. Have the authors made all data underlying the findings in their manuscript fully available (please refer to the Data Availability Statement at the start of the manuscript PDF file)?

Reviewer #1: Yes

4. Is the manuscript presented in an intelligible fashion and written in standard English?

Reviewer #1: Yes

Reviewer #1: A. Strengths:

1.The study addresses a clinically significant problem: identifying patients at risk of POD using simple, scalable tools.

2. The use of BAMCOG, a gamified cognitive screener, is innovative and relevant in the digital health space.

3. Methodology is strong: use of multiple diagnostic tools (DOS, DeltaScan), multiple cognitive tests, and validated statistical approaches.

4. Ethics and data availability are appropriately addressed.

B. Suggestions for improvement:

1. Expand discussion on limitations of DeltaScan and DOS as non-gold standard tools. While practical, their diagnostic sensitivity and specificity can vary.

2.Clarify whether BAMCOG results were adjusted for confounders like educational level or digital literacy—factors that can impact game-based performance.

3. The sample size, while justified, limits generalizability. The authors mention this, but future power calculations could be more detailed.

4. Add brief detail on usability: The manuscript states the usability study is submitted elsewhere. Consider including a summary statistic or SUS score in this paper.

5. A table summarizing key results (AUROC, CI, p-values) would help readers quickly interpret performance of BAMCOG vs MoCA.

C. Minor Edits:

1. Typo: “Postoperative Delirium(N=36)” → add a space.

2. Abbreviations like “IE” for alcohol units should be clearly defined on first use.

**Do you want your identity to be public for this peer review?** For information about this choice, including consent withdrawal, please see our Privacy Policy

Reviewer #1: No

**Figure resubmission:****Reproducibility:** To enhance the reproducibility of your results, we recommend that authors of applicable studies deposit laboratory protocols in protocols.io, where a protocol can be assigned its own identifier (DOI) such that it can be cited independently in the future. Additionally, PLOS ONE offers an option to publish peer-reviewed clinical study protocols. Read more information on sharing protocols at https://plos.org/protocols?utm_medium=editorial-email&utm_source=authorletters&utm_campaign=protocols

---

## [Editor Report · Decision Letter 1]

21 Aug 2025

Evaluating the predictive accuracy of cognitive screeners BAMCOG and MoCA in identifying postoperative delirium risk in aortic valve replacement patients: A cohort study

PDIG-D-25-00234R1

Dear Dr. te Pas,

We're pleased to inform you that your manuscript has been judged scientifically suitable for publication and will be formally accepted for publication once it meets all outstanding technical requirements.

Within one week, you'll receive an e-mail detailing the required amendments. When these have been addressed, you'll receive a formal acceptance letter and your manuscript will be scheduled for publication.

An invoice for payment will follow shortly after the formal acceptance. To ensure an efficient process, please log into Editorial Manager at https://www.editorialmanager.com/pdig/ click the 'Update My Information' link at the top of the page, and double check that your user information is up-to-date. For billing related questions, please contact billing support at https://plos.my.site.com/s/.

Kind regards,

Peter H Charlton, MEng, PhD

Section Editor

PLOS Digital Health

Additional Editor Comments (optional):

Thank you for addressing all of the comments - and for a very interesting submission.